# Little fast, little slow, should I stay or should I go? Adapting cognitive control to local-global temporal prediction across typical development

Fiorella Del Popolo Cristaldi[1‡], Lisa Toffoli[1‡]*, Gian Marco Duma[2], Giovanni Mento[1,3]

**1** NeuroDev Lab, Department of General Psychology, University of Padua, Padua, Italy, **2** Epilepsy and Clinical Neurophysiology Unit, IRCCS "E. Medea" Conegliano, Treviso, Italy, **3** Padua Neuroscience Center (PNC), University of Padua, Padua, Italy

‡ FDPC and LT contributed equally to this work and should be considered joint first authors.
* lisa.toffoli@phd.unipd.it

**Data Availability Statement:** Data and analysis code reported in this paper have been made publicly available Open Science Framework (OSF)

## Abstract

Adaptive cognitive control (CC), the ability to adjust goal-directed behavior according to changing environmental demand, can be instantiated bottom-up by implicit knowledge, including temporal predictability of task-relevant events. In S1-S2 tasks, either local (trial-by-trial hazard expectation) or global (block-by-block expectation) temporal information can induce prediction, allowing for proactive action control. Recent developmental evidence showed that adaptive CC based on global temporal prediction emerges earlier than when it is based on the local one only. However, very little is known about how children learn to dynamically adjust behavior on the fly according to changing global predictive information. Addressing this issue is nevertheless crucial to unravel the mechanisms underlying adaptive CC flexibility. Here we used a modified version of the Dynamic Temporal Prediction task to investigate how typically developing younger (6–8 years) and older children (9–11 years), adolescents (12–15 years) and adults (21–31 years) use global prediction to shape adaptive CC over time. Specifically, the short-long percentage of S2 preparatory intervals was manipulated list-wide to create a slow-fast-slow-fast fixed block sequence and test how efficiently the response speed adapted accordingly. Overall, results revealed that in all groups behavioral performance is successfully adjusted as a function of global prediction in the late phase of the task (block 3 to 4). Remarkably, only adolescents and adults exhibit an early adaptation of adaptive CC (block 1 to 2), while children younger than 11 show sluggish ability in inferring implicit changes in global predictive rules. This age-related dissociation suggests that, although being present from an early age, adaptive CC based on global predictive information needs more developmental space to become flexible in an efficient way. In the light of a neuroconstructivist approach, we suggest that bottom-up driven implicit flexibility may represent a key prerequisite for the development of efficient explicit cognitive control

and can be accessed at https://osf.io/kj94s/ DOI:
10.17605/OSF.IO/KJ94S.

**Funding:** The author(s) received no specific
funding for this work.

**Competing interests:** The authors have declared
that no competing interests exist.

## Introduction

The ability to predict the temporal occurrence of goal-relevant events is crucial for human adaptation. Previous research has demonstrated that temporally complex behaviors can be accomplished since the human brain shows intrinsic predictive properties [1, 2]. In fact, already at birth [3] or even before the expected birth time [4], the brain automatically infers statistical regularities from the environment, such as temporal regularities, to build expectancy toward relevant future events. Crucially, temporal prediction allows to proactively adjust behaviour biasing action control by regulating inhibitory and excitatory mechanisms as a function of task demand [5–15]. Given the core adaptive value of this ability, the literature has long investigated implicit temporal expectancy in newborns and children. Several studies revealed that the temporal predictions based on narrow transitional probabilities (i.e., local temporal prediction) are already in place in newborns [4, 16–21]. Differently the ability to extract higher-level, non adjacent rules (i.e., global temporal prediction) seems to emerge later on [22, 23]. Hence, local-feature based cognition may be a building block for more complex mental processes, such as the ability to handle the global gestaltic pattern of sensory stimuli.

In the context of experimental psychology, local and global temporal regularities can be manipulated to induce stimulus prediction. Specifically, in traditional S1-S2 tasks, characterized by a signal warning for an upcoming imperative stimulus, local prediction deploys within each single trial and conforms to the hazard function, so that people are faster to detect long-expected than short-expected targets [8, 24, 25]. Differently, global prediction does not depend on single trials but it requires longer accumulation of statistical evidence about the existence of general temporal patterns and any dynamic changes of these [26–29]. For example, increasing the proportion of short-expectancy trials in a single block speeds up the global stimulus presentation rate, turning into faster response speed for "fast" blocks as compared to "slow blocks", which include long trials mostly [27].

Most developmental studies have focused on the ability to use local temporal prediction to adapt response speed. They showed that that the ability to adjust response speed on the basis of local prediction is well-established at 5 years of age [28, 30, 31], and it has stable developmental trajectories until adult age in both typical [32] and atypical [33] populations. This suggests that the ability to use local temporal information to adapt cognitive control is an early and developmentally stable property of the human cognitive system. However, less is known about developmental trajectories of global temporal prediction, since the vast majority of studies used tasks with variable but equiprobable S1-S2 preparatory intervals [30–34]. In a recent study [28], we made a first step to fill this gap by using the Dynamic Temporal Prediction (DTP) task. The DTP is a novel experimental paradigm purposely designed to investigate how local (within trial) and global (between block) temporal regularities interact as two independent sources of stimulus predictability to induce adaptive cognitive control (CC). adaptive CC is defined as the ability to bottom-up extract the statistical regularities embedded in the environment to optimize behavior [35]. Noteworthy, unlike traditional cognitive control tasks (i.e., Stroop, flanker or Simon) the DTP task does not include a conflict condition, ruling out any potential confound deriving from excessive working memory load or complex instructions, as the only instruction is to press a button when the target occurs. This makes it suitable for investigating adaptive CC in young typically [28] and atypically [36–38] developing children. Nonetheless, in the context of the DTP task adaptive CC is crucial for efficient adaptation to both local and global patterns. Indeed, optimal behavioral adaptation should reflect not only the generation of adequate predictive models of events' sensory structure but also optimal flexibility in updating and implementing these predictions to proactively prepare for action by balancing excitatory and inhibitory neural mechanisms.

In a recent study [28], we showed that the children's response speed is also sensitive to the global temporal predictability, with less accurate but faster responses in fast blocks compared to slow ones. Noteworthy, unlike local prediction, global prediction becomes developmentally stable only after middle childhood [28]. This suggests that the ability to implicitly extract and use global statistical rules to adapt cognitive control may follow a prolonged ontogenetic trajectory, hence being more vulnerable to atypical development. Notably, a possible failure of local-global prediction integration may be a core deficit of neurodevelopmental disorders, including autism spectrum disorder and attention deficit/hyperactivity disorder [39, 40], Down Syndrome [38] and neurological disorders such as epilepsy [37]. In spite of this, it remains unclear how adaptive CC is flexibly shaped across time as a function of changing bottom-up predictive information manipulated block-by-block, and how development impacts on this. Indeed, while previous studies clearly reported that adaptive CC can be generated from either item-specific or list-wide manipulations already in young children [28, 41, 42], they did not track how these effects emerge and are shaped on the fly during task execution. Remarkably, the possibility to track how children extract and use global patterns dynamically (i.e., across the task) may be more informative about the in-depth, flexible nature of adaptive CC. Nonetheless, a time-on-task investigation targeting how adaptive CC emerges across-task may shed new light on *how* typically developing children learn to adjust behavior from environmentally available predictive information, paving a new way for understanding neurodevelopmental disorders characterized by a failure of predictive cognition [40].

In our previous study [28], the randomization of the blocks' presentation sequence between participants prevented from targeting block-by-block task speed changes in each age group. Hence, we could not address how adaptive CC emerges and is shaped over time by either local or global prediction. In the present study we expanded on previous research on adaptive CC by implementing a modified version of the DTP task featuring a fixed block type presentation sequence (i.e., slow-fast-slow-fast) between participants. As an important element of novelty, the fixed block type presentation sequence allowed here to provide all participants with the same global sequential transitional probabilities. This ultimately made it possible to compute how efficiently participants succeed in adapting cognitive control using changing global predictability induced bottom-up. Specifically, two different adaptive CC indexes reflecting the progressive performance adaptation in terms of response speed change (block-by-block delta) were calculated as the difference between fast and slow blocks in the first half (*early delta*), and in the second half (*late delta*) of the experiment. Finally, in order to provide a thorough developmental investigation we extended the age range to include adolescents (12–15 years old) and adults (21–31 years old). Based on previous findings [28, 41–43] we generated the following predictions. Overall, we expected to replicate previous evidence that cognitive control can be flexibly adapted on the basis of implicit local-global predictive rules since school-age [28]. Specifically, we hypothesized to find evidence of local prediction (H1a) by observing faster RTs in trials with long vs. short SOA [8, 31, 32]. As well, we expected to replicate previous evidence of global prediction (H1b) by observing faster RTs in fast vs. slow blocks respectively [27, 37, 44]. Furthermore, we expected to replicate our previous findings [28] of developmental changes in the ability to use global but not local temporal patterns to adjust behavior (H1c), as reflected by smaller delta values in adolescents and adults compared to older and younger children. Finally, as the core hypothesis of the present study, we expected that the ability to handle implicit complex temporal patterns becomes more efficient and flexible across development, although established from infancy. In other words, while children would need long exposure to statistical regularities to build up and use predictive knowledge, little accumulation of evidence would be sufficient for adolescents and adults to generate and use a global predictive model. Therefore, we speculated (H2a) that only adolescents and adults

would exhibit an early adaptation (from block 1 to block 2) of response speed during task execution. By contrast, we expected all children to need a longer accumulation of statistical evidence to implicitly learn the global temporal pattern, adapting their response speed only in the late stage of the task, that is, between block 3 and 4. This hypothesis comes from previous studies showing that cognitive flexibility, even when explicit demand for conflict adaptation is minimized, shows a prolonged developmental progression, establishing adult-like efficiency only during adolescence [45–47]. On the other hand, we speculated that a longer accumulation of evidence about the global statistical pattern would smooth out any developmental effect. Therefore, we expected (H2b) to observe an age-independent adaptation of cognitive control in the late stage of the task (from block 3 to block 4), with all participants succeeding to adapt response speed to task speed changes.

## Methods

### Participants

Two hundred and seventy-one participants were initially enrolled from local primary and secondary schools in the Venetian Region of Italy, and were divided into one of three age groups: younger children (6–8 years old), older children (9–11 years old) and adolescents (12–15 years old). Adults were the children's parents, or undergraduate psychology students at the University of Padua. All participants that reported having neurological or psychiatric disorders were excluded. All participants had normal or corrected-to-normal vision. The demographic characteristics of the four groups are described in Table 1.

### Ethics statement

All adult participants signed a written consent form, and children's parents provided written consent for their children's participation. All experimental procedures were approved by the Ethics Committee of the School of Psychology of the University of Padua (protocol no. 3666) and were conducted according to the principles expressed in the Declaration of Helsinki.

### Stimulus material and procedure

**Experimental procedure.** OpenSesame software [48] was used to create and administer the stimuli. Adult data were collected online by using the JATOS hosting server [49], an open-source web platform for online studies. Whereas, children and adolescents were tested in the University laboratory. Since a preliminary study comparing online vs. laboratory data collection with the same experimental task [50] showed no significant differences in the targeted experimental effects, we analyzed the data together. Participants were asked to sit comfortably in a chair at a viewing distance of around 60 cm from the monitor. All participants performed a modified version of the DTP task [28], a warned simple RT task purposely conceived to

**Table 1. Main demographic characteristics of the study's participants.**

| Group | Mean age ± SD (range) | Gender | | n |
| --- | --- | --- | --- | --- |
| | | Female | Male | |
| Younger children | 7.4 ± 0.7 year (6–8) | 25 | 24 | 49 |
| Older children | 10.0 ± 0.8 year (9–11) | 35 | 33 | 68 |
| Adolescents | 12.9 ± 0.9 year (12–15) | 17 | 18 | 35 |
| Adults | 23.2 ± 1.3 year (21–31) | 95 | 24 | 119 |

investigate children's behavioral performance (i.e., both accuracy and speed) in relation to local and global probabilistic rules as two distinct sources of temporal predictability.

**Trial structure.**   Each trial began with the display of a warning visual stimulus (S1), followed by the presentation of an imperative visual stimulus (S2) that stayed on the screen for a maximum of 1,500 ms. S1 consisted of a picture of a black camera lens surrounded by a circle (total size of the simulus: 840 × 840 pixels, 144 dpi). S2 consisted of a picture of a cartoon animal (in each trial, the cartoon animal was one of the following: sheep, cat, tortoise, owl, parrot, rabbit, dog), which was displayed centrally within the camera lens. The inter-trial-interval (ITI; i.e., the temporal interval between two consecutive trials) was randomly manipulated between 200 and 400 ms. The task consisted of a speeded target detection: participants were required to press the spacebar of the keyboard with the index finger of the dominant hand as quickly as possible at target occurrence. To encourage good performance in the participants, they were given the following instruction: "*Hi*! *These cute little animals are playing hide and seek in the woods*! *Your job is to take a photo of them as quickly as possible when they appear in view of your camera. You can take a photo by pressing the spacebar. Find them all*! *But take care —if you press the bar too soon or too late, they will run away*!"

**Local predictive context.**   To investigate the effect of the local predictive context on behavioral performance, the S1-S2 stimulus onset asynchrony (SOA; i.e., the temporal interval between S1 and S2 within a trial) was varied trial by trial within each experimental block so that two possible fixed foreperiod intervals were created. Unlike the original DTP version [28] here we used a simplified version only including a short (400 ms) and a long (1,000 ms) SOA within each trial, as these two foreperiods have been proved to reliably induce hazard-related, local prediction effects on RTs in children This manipulation, illustrated in Fig 1A, introduced in each block two levels of temporal preparation to S2 onset, and it allowed us to investigate local prediction as the effect of the stimulus hazard rate on task performance. In fact, the use of different S1-S2 SOA intervals is expected to dynamically bias subjective temporal expectancy [8, 24, 25, 51–53], with participants being fastest at detecting the targets occuring at the longer SOA and slowest at those appearing at the short SOA [8, 54].

**Global predictive context.**   As illustrated in Fig 1B, to assess the effect of the global changes in the predictive context, different probability distributions per each SOA foreperiod were introduced and manipulated block-wise. Specifically, in fast blocks, an a priori biased distribution toward the short SOA was delivered. In particular, the relative percentage was 70% and 30% for the short and long SOA, respectively. This kind of distribution, also known in the literature as non-aging distribution [27, 29] leads to a high stimulus presentation rate, resulting in a fast paced block.

By contrast, in slow blocks, an a priori biased distribution toward the long SOA was delivered. In particular, the relative percentage was 30% and 70% for the short and long SOA, respectively. This kind of distribution, also known in the literature as aging distribution (27,29) turns into a low stimulus presentation rate, yielding a slow paced block.

**Experimental design.**   Two experimental blocks per type were delivered in a fixed order (i.e., slow-fast-slow-fast), for a total of four blocks. This order allowed to disentangle adaptation (reduced RTs in the last fast block) and fatigue (increase RTs in the last fast block), two aspects that would have been confounded in a fast-slow-fast-slow sequence (i.e., both adaptation and fatigue would be reflected as an increase in RTs in the last slow block). Each block included 40 trials, for a total of 160 trials. The total length of the experiment was about 10 minutes. Participants were not told about the presence of between-block different probabilistic distributions to ensure they did not know about the global rule changes. Moreover, no pauses were introduced between blocks. In this way the block-by-block global changes were never implicitly suggested by task interruptions. All blocks were matched for sensorimotor

# (a) TRICL STRUCTURE

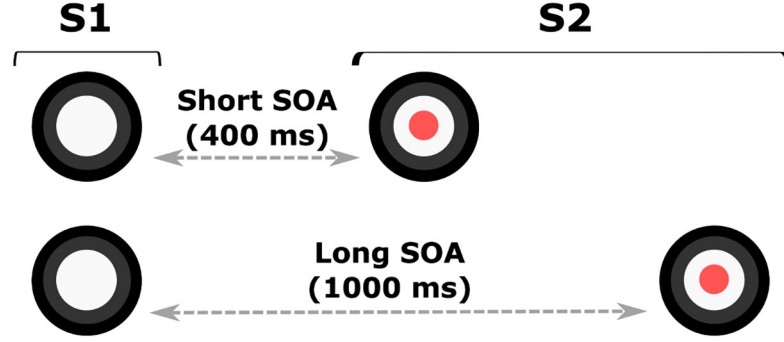

# (b) EXPERIMENTAL DESIGN

|  |  | LOCAL | |
| --- | --- | --- | --- |
|  |  | Short (400 ms) | Long (1000 ms) |
| GLOBAL | Fast Block | 70% | 30% |
| | Slow Block | 30% | 70% |

# (c) TASK DESIGN

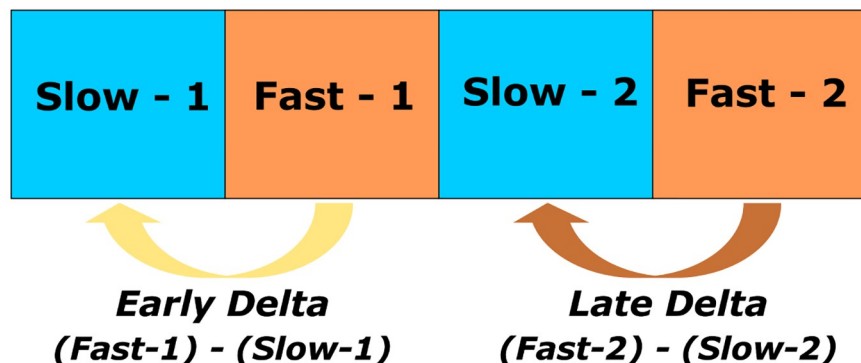

**Fig 1. Modified version of the Dynamic Temporal Prediction (DTP) task.** The DTP task was purposely designed to investigate the effect of both local and global predictive rules on implicit adaptive CC. As an important element of novelty, we introduced a fixed block-type presentation (i.e., slow-fast-slow-fast) between participants to allow the investigation of local-global flexibility of implicit temporal prediction. The circle (S1) warned children on the presentation of the imperative S2 stimulus (an animal cartoon; here represented with colored disks for illustrative purposes). Participants had to make speeded reaction times at S2 onset by pressing the space button on the keyboard. The effect of local prediction was assessed by manipulating S1-S2 stimulus onset asynchrony (SOA) within each

experimental block (a). The effect of global prediction was assessed by manipulating the between-block, a priori percentage of each SOA to create two probabilistic distributions in which the SOAs were skewed toward the short (fast block) or long (slow block) SOA (b). Finally, adaptive motor control reflecting the progressive block-by-block performance adaptation was assessed calculating two delta scores, corresponding to the difference between the first (slow-1) and the second (fast-1) block (*early* delta), and between the third (slow-2) and the fourth (fast-2) block (*late* delta) (c).

requirements, as the visual stimuli and the required response were always the same across the experiment. The only differences were related to the changes in the predictive context experienced through the task (namely, the different block-by-block global predictive ratios). Before starting the experimental session, participants were presented with a block of 10 practice trials to ensure they understood task instructions. During practice each participant received trial-by-trial feedback based on their performance. Specifically, a puzzled face was displayed in cases in which anticipatory (before target onset) or premature ($<$ 150 ms after target onset) responses were provided. A sleepy face was displayed if the RT was between 800 and 1,500 ms from target onset. Finally, a smiley face was displayed if the response was correct, that is, when it occurred between 150 and 800 ms. No feedback was delivered during the experimental session.

**Data analysis.**   The study has a 4 (*age group*, between-subjects: adults 21–31 years old, adolescents 12–15 years old, older children 9–11 years old, younger children 6–8 years old) × 4 (*block sequence*, within-subjects: slow-1, fast-1, slow-2, fast-2) × 2 (*SOA*, within-subjects: short, long) mixed design. Univariate outliers on RTs and accuracy were detected through Median Absolute Deviation values (MAD $>$ 3) (R package: Routliers) [55]. We detected and discarded 5 univariate outliers (4 children, 1 adult). A final sample of 266 participants (148 children and 118 adults) was included into data analysis.

Before performing the analyses, we pre-processed RTs according to the following steps: (i) trimming RTs between 150 and 1,000 ms [56]; (ii) discarding RTs of incorrect trials; (iii) adjusting RTs for the speed-accuracy trade off by means of Inverse Efficiency Score (IES) transformation [57]; (iv) log-transforming IES to account for their skewed distribution [56, 58]. Furthermore, we computed early and late delta scores separately per SOA intervals as the difference in RTs between the second (fast-1) and the first (slow-1), and between the fourth (fast-2) and the third (slow-2) blocks, respectively (Fig 1C).

In order to test our hypotheses (H1a, H1b, H2a, H2b), for each dependent variable (DV) we fitted the following *Linear Mixed-effect Models* (LMMs) or *Linear Models* (LMs) (R packages: lme4, car) [59, 60]:

1. Log-transformed IES: LMMs with *age group*, *block sequence*, *SOA* and their interactions as fixed factors, and random intercept for participant;

2. Early and late delta scores: LMs with *age group*, *SOA* and their interactions as predictors.

LMMs and LMs effects were evaluated using *F*-test and *p*-values, calculated either via Satterthwaite's degrees of freedom method ($\alpha$ = .05, R package: lmerTest) [61], or Type III Analysis of Variance (R package: car) [60], respectively. Post-hoc pairwise comparisons between the levels of fixed factors were tested by means of estimated marginal means (EMMs) contrasts, Tukey adjusted for multiple comparisons (R package: emmeans) [62]. For each model we reported the estimates with standard error (*SE*), 95% confidence interval (*CI*), and the associated statistics (*t*-test). Parameters are reported in the logit scale for models on IES. Moreover, for each LMM we reported the marginal and conditional $R^2$ (estimated as in [63]), and for each LM we reported the $R^2$ and adjusted $R^2$.

## Results

### Accuracy

In all age groups mean accuracy was above 97%, thus we chose not to analyze it because it reached a ceiling level.

### Inverse efficiency score

The LMM on log-transformed IES is summarized in Fig 2 and S1 Table.

We found significant main effects of *age group* ($F(3, 263) = 35.59$, $p < .001$), *block sequence* ($F(3, 40879) = 195.91$, $p < .001$), and *SOA* ($F(1, 40878) = 1781.69$, $p < .001$). As for the *age group* main effect, we found expected evidence of increasingly faster RTs, as reflected by smaller IESs, with increasing age, regardless of block and SOA (see S2 Table for post-hoc comparisons). The *block sequence* main effect showed evidence of a progressive slow down of RTs along the task, regardless of SOA and age group, confirming previous evidence of global prediction on response speed [28] (see S3 Table for post-hoc comparisons). As hypothesized on the *SOA* main effect (H1a), we also found faster RTs in trials with long as compared to short SOA (short vs. long: 0.12, $SE = 0.003$, $t(40878) = 42.21$, $p < .001$), regardless of block and age group.

Moreover, we found a significant *age group* × *block sequence* × *SOA* interaction ($F(9, 40878) = 6.29$, $p < .001$). This effect confirmed the hypothesis that behavioural performance can be proactively shaped by implicit changes in global temporal pattern (H1b). Indeed, post-hoc comparisons yielded evidence that only in trials with short SOA (see Table 2) adults showed a flexible block-by-block RTs adaptation, displaying significantly faster RTs (and thus, showing decreasing IESs) as the task speeded up. Indeed, participants became faster from the first (slow-1) to the second (fast-1), and from the third (slow-2) to the fourth (fast-2) block. As expected on the basis of the global properties of the task, we also found slower RTs (and thus, increasing IESs) as the task slowed down, that is, from the second (fast-1) to the third (slow-2) block. In adolescents, the block-by-block RTs trend in trials with short SOA was similar to adults, although only the comparison between the second (fast-1) and the third (slow-2) block reached statistical significance. In both older and younger children, instead, the block-by-block RTs trend in trials with short SOA showed significant and progressive slower RTs from the first (slow-1) to the second (fast-1) and third (slow-2) blocks, followed by slightly faster RTs from the third (slow-2) to the fourth (fast-2) block, which however did not reach statistical significance. Unlike short-SOA trials, in the case of long preparatory intervals (long-SOA trials), block-by-block RTs trends showed progressively slower RTs in all participants (see S4 Table for post-hoc comparisons).

### Behavioral adaptation across the task

The LMs on early and late delta scores as two measures of response speed adaptation are summarized in Fig 3 and S5 Table.

**Early delta scores.** From the LM on early delta scores significant main effects of *age group* ($F(3, 41164) = 919.39$, $p < .001$) and *SOA* ($F(1, 41164) = 45.10$, $p < .001$) emerged, better specified by a significant *age group* × *SOA* interaction ($F(3, 41164) = 471.92$, $p < .001$).

As for the *age group* main effect, as expected post-hoc contrasts (see S6 Table) yielded evidence of smaller early delta scores with increasing age, suggesting a progressively more efficient adaptation to the speed changes of the first half of the task. As for the *SOA* main effect, we found smaller early delta scores in trials with short as compared to long SOA intervals (short vs. long: -3.78, $SE = 0.56$, $t(41164) = -6.72$, $p < .001$).

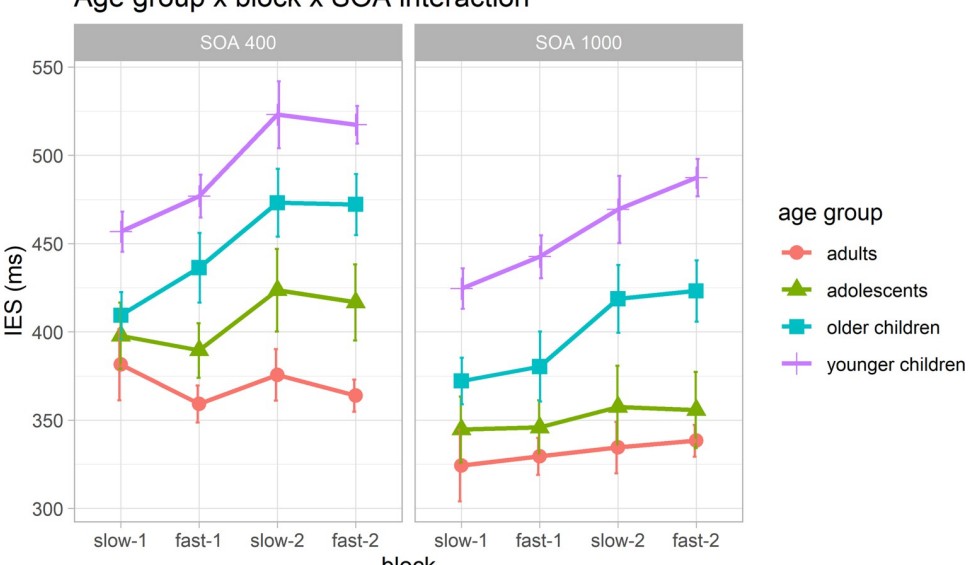

**Fig 2. Age group × block × SOA interaction plot on Inverse Efficiency Score (IES) (ms).** The panels show the block-by-block changes in response speed for short (left panel) and long (right panel) preparatory intervals (SOA). Only for short SOA trials we found an age-dependent motor adaptation, with adults and adolescents showing faster RTs as the task speeded up. In the case of long-SOA trials all participants showed a general block-by-block RTs trend with progressively slower RTs regardless of the global predictive context (block-type). Error bars represent standard error (*SE*).

Post-hoc contrasts decomposing the *age group × SOA* interaction showed that in trials with short SOA intervals (see Table 3), as hypothesized (H2b), we found an age-dependent effect of the behavioral adaptation, with older groups performing better than the younger ones. Namely, in trials with short SOA, the mean early delta values were negative in adults and adolescents, but positive in older and younger children (adults: -22.97 ms; adolescents: -8.91 ms; older children: 26.15 ms; younger children: 22.66 ms). This suggested that in those trials adults

**Table 2. Post-hoc contrasts of the *age group × block sequence × SOA* interaction effect of the IES model.**

| SOA | age group | contrast | estimate | SE | df | t | p |
|-----|-----------|----------|---------|-----|-----|-----|-----|
| **400** | adults | slow-1 vs. fast-1 | 0.054 | 0.008 | 40880 | 7.158 | < .001 |
| | | fast-1 vs. slow-2 | -0.042 | 0.008 | 40878 | -5.600 | < .001 |
| | | slow-2 vs. fast-2 | 0.025 | 0.008 | 40880 | 3.330 | **.005** |
| | adolescents | slow-1 vs. fast-1 | 0.025 | 0.014 | 40878 | 1.808 | .269 |
| | | fast-1 vs. slow-2 | -0.088 | 0.014 | 40878 | -6.273 | < .001 |
| | | slow-2 vs. fast-2 | 0.027 | 0.014 | 40878 | 1.940 | .211 |
| | older children | slow-1 vs. fast-1 | -0.054 | 0.010 | 40878 | -5.334 | < .001 |
| | | fast-1 vs. slow-2 | -0.084 | 0.010 | 40878 | -8.218 | < .001 |
| | | slow-2 vs. fast-2 | 0.006 | 0.010 | 40878 | 0.550 | .947 |
| | younger children | slow-1 vs. fast-1 | -0.044 | 0.012 | 40878 | -3.630 | **.002** |
| | | fast-1 vs. slow-2 | -0.094 | 0.012 | 40879 | -7.672 | < .001 |
| | | slow-2 vs. fast-2 | 0.027 | 0.012 | 40879 | 2.164 | .133 |

Only contrasts relative to short SOA intervals are reported. For each contrast, we report the estimate (in logit scale), standard errors (*SE*), degrees of freedom (*df*), and the associated statistic (*t*-test).

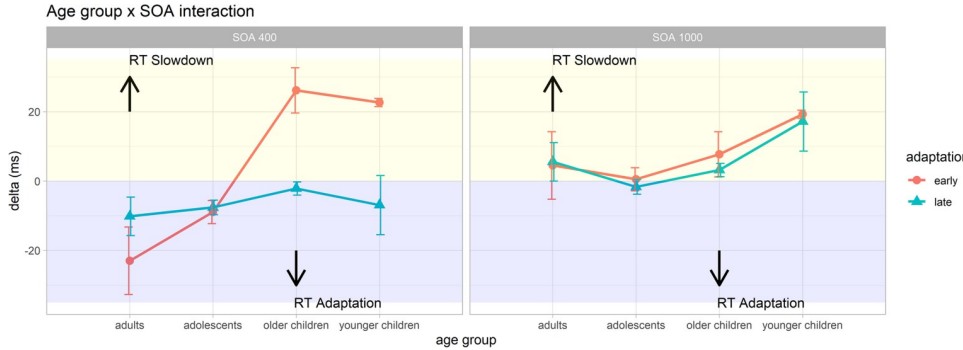

**Fig 3. Early and late behavioral adaptation.** The panels represent the developmental trajectories of the ability to adapt response speed on global prediction changes relatively to the early (block 2 *minus* block 1; red line) and late (block 4 *minus* block 3; light blue line) stages of the task and separately for short and long preparatory intervals (SOA). The left panel (short SOA) shows negative delta scores (RT adaptation; violet half) only for adolescents and adults and positive delta scores for (RT slowdown; yellow half) children. Conversely, all participants exhibit RT slowdown with an increased preparatory interval (long SOA). Error bars represent standard error (*SE*).

and adolescents actually displayed faster RTs from the first (slow-1) to the second (fast-1) block, whereas older and younger children displayed slower RTs. In trials with long SOA intervals (see S7 Table), instead, both adults and adolescents showed smaller early delta scores than older and younger children; adults showed greater scores than adolescents; and older children showed smaller scores than younger children. Most interestingly, in the long SOA condition all age groups showed positive mean early delta values (adults: 4.52 ms; adolescents: 0.54 ms; older children: 7.71 ms; younger children: 19.28 ms), indicating a general RTs slowdown from the first (slow-1) to the second (fast-1) block. In other words, the delta scores indicating the ability to efficiently regulate behavior already during the early stage of the task (from block one to block two) clearly revealed a developmental trend, with good adaptation only after adolescence.

**Late delta scores.** In the LM on late delta scores, we found significant main effects of *age group* ($F(3, 41164) = 53.03$, $p < .001$) and *SOA* ($F(1, 41164) = 555.53$, $p < .001$), better specified by a significant *age group* × *SOA* interaction ($F(3, 41164) = 63.85$, $p < .001$).

**Table 3. Post-hoc contrasts of the *age group* × *SOA* interaction effect of the early and late delta scores models.**

| index | SOA | contrast | estimate | SE | df | t | p |
|---|---|---|---|---|---|---|---|
| early delta | 400 | adults vs. adolescents | -14.05 | 1.133 | 41164 | -12.40 | < **.001** |
| | | adults vs. older children | -49.11 | 0.901 | 41164 | -54.48 | < **.001** |
| | | adults vs. younger children | -45.63 | 1.016 | 41164 | -44.92 | < **.001** |
| | | adolescents vs. older children | -35.06 | 1.230 | 41164 | -28.51 | < **.001** |
| | | adolescents vs. younger children | -31.58 | 1.316 | 41164 | -23.99 | < **.001** |
| | | older children vs. younger children | 3.48 | 1.123 | 41164 | 3.10 | **.010** |
| late delta | 400 | adults vs. adolescents | -2.56 | 1.090 | 41164 | -2.346 | .088 |
| | | adults vs. older children | -8.03 | 0.868 | 41164 | -9.257 | < **.001** |
| | | adults vs. younger children | -3.26 | 0.978 | 41164 | -3.332 | **.005** |
| | | adolescents vs. older children | -5.47 | 1.184 | 41164 | -4.624 | < **.001** |
| | | adolescents vs. younger children | -0.70 | 1.267 | 41164 | -0.553 | .946 |
| | | older children vs. younger children | 4.77 | 1.081 | 41164 | 4.416 | < **.001** |

Only contrasts relative to short SOA intervals are reported. For each contrast, we report the estimate (in ms), standard errors (*SE*), degrees of freedom (*df*), and the associated statistic (*t*-test).

As for the *age group* main effect, as expected post-hoc contrasts (see S6 Table) yielded evidence of increasingly smaller late delta scores with increasing age, suggesting a progressively more efficient adaptation to the speed changes of the second half of the task, with the exception of adults vs. adolescents comparison showing more negative delta scores in the latter group. As for the *SOA* main effect, we found smaller late delta scores in trials with short as compared to long SOA intervals (short vs. long: -12.77, *SE* = 0.54, *t*(41164) = -23.57, *p* < .001).

Post-hoc contrasts decomposing the *age group* × *SOA* interaction showed that in trials with short SOA intervals (see Table 3) as hypothesized (H2a) adults and adolescents showed smaller late delta scores than the other age groups, while older children showed greater late delta scores than younger children. Interestingly, in trials with short SOA, mean late delta values were negative in all age clusters (adults: -10.16 ms; adolescents: -7.60 ms; older children: -2.13 ms; younger children: -6.90 ms), suggesting that in those trials all age groups actually displayed faster RTs from the third (slow-2) to the fourth (fast-2) block. In trials with long SOA intervals (see S7 Table), instead, adults showed greater late delta scores than both adolescents and older children, and smaller scores than younger children; adolescents showed smaller scores than older and younger children; and older children showed smaller scores than younger children. In the long SOA condition, all age groups except adolescents showed positive mean late delta values (adults: 5.56 ms; adolescents: -1.66 ms; old children: 3.21 ms; young children: 17.18 ms), indicating a RTs slowdown from the third (slow-2) to the fourth (fast-2) block.

## Discussion

In this study, we investigated how adaptive cognitive control (CC) is dynamically shaped as a function of changing bottom-up predictive information across typical development (6–15 years old) and adulthood (21–31 years old). To this purpose we used a modified version of the DTP task [28] in which temporal prediction toward task-relevant stimuli is generated on the basis of both local (within trial expectancy) and global (between block expectancy) probabilistic rules. Noteworthy, we used a fixed (i.e., slow-fast-slow-fast) block-type sequence to track how efficiently behavioral performance adheres to bottom-up fixed global predictive information by taking a developmental perspective.

### Local predictive context (H1a)

In line with our first hypothesis (H1a), results revealed that all participants were overall faster when the target was preceded by long as compared to short SOA intervals. This phenomenon conforms to the hazard function distribution and is experimentally reflected by the well-known foreperiod effect. According to this effect, a speed advantage occurs following long versus short preparatory intervals within trials, due to the unidirectional flow of time that biases target predictability intrinsically [8, 10, 24, 25, 52]. These results nicely align with previous studies suggesting the presence of early, stable developmental trajectories of the ability to use local probabilistic rules to establish implicit temporal preparation [28, 31–33].

### Global predictive context (H1b)

As regards global rule changes, we found a speed advantage in fast blocks among adults and adolescents but only in trials with short SOA intervals. Noteworthy, these results only partially replicate previous results, in which it emerged a global speed advantage in target detection in fast blocks [28]. Specifically, our original findings showed that the ability to extract global probabilistic rules is already in place from 5 years of age and becomes stable after the age of 7. Instead, here we report a global effect only from early adolescence on. However, when comparing these findings, one should consider that the global temporal expectancy induced in our

original study was probably easier to catch since the proportion between long and short SOA intervals was more unbalanced compared to the one employed in the present study (respectively, 60–8% vs 70–30%). This suggests that younger and older children might be able to extract global predictive rules only under certain circumstances, such as when these can be easily extracted from the context.

Our findings confirm the presence of distinct developmental patterns of local and global stimulus predictability on adaptive CC [28]. Also, they further support that predictions based on local rules ontogenetically precede the ability to efficiently use global patterns [64, 65]. The evidence that bottom-up knowledge, such as local-global temporal regularity, may shape adaptive CC is in line with a recent theoretical proposal known as 'Learning Perspective on Cognitive Flexibility' [35], suggesting that implicit processes such as associative learning might actually be functional in achieving cognitive control. For instance, it has been shown that subliminal statistical regularities in stimulus-response or in response-reward association may shape cognitive control by reducing the switch cost (see [66] for a review) engaged in shifting between different task sets. In other words, the concept of 'being flexible' may prescind from the representational format (i.e., explicit vs. implicit) required by the task at hand. The idea of an 'implicit cognitive control' has been also recently corroborated by experimental data [36, 37, 41, 42, 44, 67]; for a review, see [68].

### Flexible behavior adaptation (H2a and H2b)

The main novelty introduced by the present DTP version was the fixed block type sequence presentation. As expected, participants showed a progressive slowing down of response speed across the task. This is suggestive of a general fatigue and/or sustained attentional decline, as typically found in speeded target detection tasks [69]. However, when looking at block-by-block response speed changes separately by preparatory interval (i.e., short- vs. long-SOA), we found that only adolescents and adults conformed to what was expected on the basis of the global predictive context. Specifically, the fixed sequence alternating slow-fast-slow-fast blocks was expected to counterbalance the overall speed reduction across the task due to fatigue. In line with this assumption, in adults and adolescents we observed that performance mirrored the global temporal pattern of the task. In other words, in trials with short SOA, RTs speeded up as the task speeded up while they slowed down as the task slowed down. Differently, in trials with long SOA intervals participants of all ages did not show behavioral adaptation. They exhibited indeed a progressive slowdown in response times, in line with what would be expected due to normal fatigue. The different block-by-block response speed pattern observed for short and long SOA may be due to more than one factor. First, it is worth noting that long SOA intervals are relatively rare in fast blocks. Hence, participants may be less prepared to detect them, resulting in slower RTs. In other words, the more frequent the SOA, the faster the response speed relative to that SOA. A second reason for short-long SOA differences may be due to neurofunctional aspects. Indeed, there is evidence demonstrating that interval timing works differently for sub- or supra-second durations [70]. In particular, one hypothesis argues that sub-second intervals may be automatically processed by subcortical structures, such as basal ganglia and the cerebellum, while supra-second durations may be more easily represented in cortical regions, including primary and associative areas [see 71 for a review]. The reason for this is that long durations engage controlled timing processes and, as such, would need a higher load of working memory and attention in order to be encoded, maintained and retrieved. Interestingly, empirical evidence has pointed out that basal ganglia are strongly involved in implicit learning tasks [72, 73], including those requiring to update associative learning according to new incoming information [e.g., reversal learning; 74]. This may explain

why only in short-SOA trials participants exhibited flexibility in learning and updating implicit global temporal patterns. By contrast, the need for greater cognitive resources in long-SOA trials may have called for more distributed cortical activity, potentially shadowing any effect of temporal predictability. This hypothesis is undoubtedly promising and offers hints to bridge behavioral and neural research to understand adaptive CC development. However, it remains speculative and needs further empirical evidence to be confirmed by future studies. Beside this, our results suggest that the global stimulus predictability given by the block type presentation sequence (i.e., slow-fast-slow-fast) succeeded in inducing adaptive CC by shaping on the fly participants' performance among adolescents and adults.

In order to better isolate age-related differences in block-by-block speed adaptation, we computed *early* and *late* delta scores separately by SOA. The *early* delta represents an index of the ability to implicitly extract and use global probabilistic rules at an early stage of the task (i.e., block 1 to 2). Conversely, the *late* delta represents an index of behavioral adaptation at a late stage of the task (i.e., block 3 to 4). Since faster RTs are expected to be elicited in fast as compared to slow blocks [27, 28, 37, 44], an effective and flexible block-by-block speed adaptation should be reflected by a negative delta score.

As expected, results showed progressively smaller *early* delta values with increasing age, suggesting a developmental refinement of the capacity to flexibly adapt behavior to complex statistical patterns. More specifically, adults and adolescents showed negative *early* delta values only in trials with short SOA intervals; whereas, older and younger children showed positive *early* delta values. By contrast, as regard to long SOA trials, all participants showed positive *early* delta values, suggesting that all age groups reduced response speed from the first to the second block. This trend supports the hypothesis H2b, since it suggests a developmental dissociation in the ability to flexibly extract and use global temporal rules. In fact, only adults and adolescents promptly adapted their behavioral performance between the first two blocks of the task, while children younger than 11 did not.

By contrast, concerning the behavioral adaptation in the late stage of the task, results showed progressively smaller *late* delta values with increasing age, even in this case only in trials with short SOA intervals. Interestingly, in trials with short SOA, the *late* delta value was negative across all groups, while in trials with long SOA intervals, adults showed greater *late* delta values compared to adolescents and older children. Moreover, only adolescents showed negative *late* delta value. This may reflect a greater adherence of adults to the global SOA distribution within the block (slower responses toward long SOA intervals since they are less frequent in fast blocks).

According to the 'Flexible Control Model' [75, 76], a possible explanation for this developmental dissociation may rely on the ability to efficiently update internal predictive models on the basis of the variability inferred from the context. Following this reasoning, it is possible that below 11 years of age children experience more effort and need longer time to efficiently carry out this process. However, in the second part of the task, adaptation occurred across all groups. This suggests that children aged 6 to 11 are able to update and use an internal predictive model according to the global temporal structure once the accumulated, incoming sensory evidence is robust enough. In other words, they need more time to succeed shaping their behavior based on the internally-generated model, and to adjust cognitive control consequently. Another possible explanation may involve a de-synchronization between internal predictive models and the ability to use these models to efficiently shape action preparation. Indeed, one study suggested that hierarchically-nested levels of temporal predictability recruit different oscillatory neural mechanisms that must be coupled to optimize motor control in children [77]. Hence, we could speculate that only from adolescence onward the neural circuits connecting both cortical and sub-cortical structures underpinning globally-induced implicit

predictive cognition (i.e., generation, implementation and updating) with action control (i.e., executing or inhibiting a response) become functionally efficient enough to allow flexible behavioral adaptation to constantly changing environmental requests.

## Conclusions

In the present study we provide evidence that contextual characteristics (i.e., implicit local-global predictive rules) can induce adaptive CC by allowing flexible behavioral adaptation across development. In the light of the present results, we speculate that adaptive CC could be a developmental precursor for flexibly adapting behavior when it is explicitly required. In fact, the exploitation of contextual regularities through implicit learning might be considered as an interface between the human cognitive system and the environment. The better the interface bottom-up capitalizes our cognitive resources on the basis of interiorized-statistical models, the better our behavioral adaptation. The ability to flexibly shift from local to global predictive patterns can be considered as a domain-general feature characterizing implicit learning across childhood. Therefore, the use of a dynamic local-global predictive context, as the one elicited in the present DTP task version, might provide a better dimensional characterization of different atypical profiles on the basis of local-global flexibility of implicit predictive cognition. Noteworthy, this implies important clinical implications such as the possible development of early screening tools for the identification of at-risk developmental trajectories and neuropsychological endophenotypes.

One main limitation of this study is the presence of unequal sample sizes per age groups. Therefore, we ensured that every sample size was large enough to afford statistical reliability and modelled the inter-individual variability during data analysis [78].

Future research may expand upon the presented results investigating how our behavior flexibly adapts to local-global predictive rules across lifespan.

## Supporting information

**S1 Table. Results of LMM on log-transformed IES.** We report the unstandardized regression coefficients (in logit scale), standard errors (*SE*), 95% confidence intervals (*CI*), degrees of freedom (*df*), and the associated statistic (*t*-test).
(DOCX)

**S2 Table. Post-hoc contrasts of the *age group* main effect of the IES model.** For each contrast, we report the estimate (in logit scale), standard errors (*SE*), degrees of freedom (*df*), and the associated statistic (*t*-test).
(DOCX)

**S3 Table. Post-hoc contrasts of the *block sequence* main effect of the IES model.** For each contrast, we report the estimate (in logit scale), standard errors (*SE*), degrees of freedom (*df*), and the associated statistic (*t*-test).
(DOCX)

**S4 Table. Post-hoc contrasts of the *age group* × *block sequence* × *SOA* interaction effect of the IES model.** Only contrasts relative to long SOA intervals are reported. For each contrast, we report the estimate (in logit scale), standard errors (*SE*), degrees of freedom (*df*), and the associated statistic (*t*-test).
(DOCX)

**S5 Table. Results of LMs on early and late delta scores.** For each model, we reported the unstandardized regression coefficients, standard errors (*SE*), 95% confidence intervals (*CI*),

degrees of freedom (*df*), and the associated statistic (*t*-test).
(DOCX)

**S6 Table. Post-hoc contrasts of the *age group* main effect of the early and late delta scores models.** For each contrast, we report the estimate (in ms), standard errors (*SE*), degrees of freedom (*df*), and the associated statistic (*t*-test).
(DOCX)

**S7 Table. Post-hoc contrasts of the *age group* × *SOA* interaction effect of the early and late delta scores models.** Only contrasts relative to long SOA intervals are reported. For each contrast, we report the estimate (in ms), standard errors (*SE*), degrees of freedom (*df*), and the associated statistic (*t*-test).
(DOCX)

## Acknowledgments

The authors wish to thank all the children who participated in the study and their families. A special thanks goes to Dr. Irene Coppini for helping with data collection, and to all the staff at the Istituto Comprensivo "Bonaccorso da Montemagno" in Quarrata (Prato, Italy). We also thank Dr. G. Mellori and Prof. F. Bulsara for their precious suggestions.

## Author Contributions

**Conceptualization:** Gian Marco Duma, Giovanni Mento.

**Data curation:** Fiorella Del Popolo Cristaldi.

**Formal analysis:** Fiorella Del Popolo Cristaldi.

**Funding acquisition:** Giovanni Mento.

**Investigation:** Giovanni Mento.

**Methodology:** Gian Marco Duma, Giovanni Mento.

**Project administration:** Giovanni Mento.

**Resources:** Giovanni Mento.

**Supervision:** Giovanni Mento.

**Visualization:** Fiorella Del Popolo Cristaldi.

**Writing – original draft:** Fiorella Del Popolo Cristaldi, Lisa Toffoli, Giovanni Mento.

**Writing – review & editing:** Gian Marco Duma, Giovanni Mento.

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
