## [Decision Letter · Decision Letter 0]

4 Jan 2023

PONE-D-22-33019

Little fast, little slow, should I stay or should I go? Adapting cognitive control to local-global temporal prediction across typical development

PLOS ONE

Dear Dr. TOFFOLI,

Thank you for submitting your manuscript to PLOS ONE. After careful consideration, we feel that your manuscript has merit but does not fully meet PLOS ONE’s publication criteria as it currently stands. Therefore, we invite you to submit a revised version of the manuscript that addresses the points raised by the reviewers during the review process.

The reviewers made positive comments about your paper, but they raised some concerns that need to be addressed before publication. In particular, I agree with Reviewer #2 that suggests to clarify some aspects of the Methods section.  

Overall, I think that  the manuscript is well written, the study is well designed and that the topic is interesting.

We look forward to receiving your revised manuscript.

Kind regards,

Valentina Bruno

Academic Editor

PLOS ONE

Journal Requirements:

Reviewers' comments:

Reviewer's Responses to Questions

Comments to the Author

1. Is the manuscript technically sound, and do the data support the conclusions?

Reviewer #1: Yes

Reviewer #2: Yes

2. Has the statistical analysis been performed appropriately and rigorously?

Reviewer #1: Yes

Reviewer #2: Yes

3. Have the authors made all data underlying the findings in their manuscript fully available?

Reviewer #1: Yes

Reviewer #2: Yes

4. Is the manuscript presented in an intelligible fashion and written in standard English?

Reviewer #1: Yes

Reviewer #2: Yes

5. Review Comments to the Author

Reviewer #1: General comments:

The authors presented a fixed block type presentation sequence to participants to assess how successful they were in adapting their cognitive control when exposed to changing global predictability, that was induced in a bottom-up way. The fixed presentation allowed a direct comparison across four age-groups. They used a time-on-task method to measure early and late changes in response time across the course of the study. The aims and the hypotheses were clear. There were a good number of participants in each of the four age groups. Two stimulus onset asynchrony intervals were used – 400ms and 1000ms. The task was appropriate and sounded quite fun and engaging for children. The data were treated appropriately. The statistics used were excellent. The discussion was very clearly written. The results were discussed appropriately in light of previous research and theory, and potential links with brain mechanisms were suggested. Figure 3 was particularly helpful.

Specific comments:

1. Discussion section – are there any limitations the authors would like to note?

Reviewer #2: I have a few minor comments and questions that I'd like to see addressed in a revision (listed in chronological order):

- Abbreviation ACC for adaptive cognitive control: I recommend against using this abbreviation because it is already a common abbreviation for a brain area (anterior cingulate cortex). This could be confusing for some readers.

- Participants: There are very unequal sample sizes per age group (range 35-119). Is there a reason for this?

- Trial structure: The authors write "The inter-trial-interval (ITI) was randomly manipulated between 200 and 400 ms", but later on it reads "her we used a simplified verision only including a short (400 ms) and a long (1,000 ms) SOA interval". It took me a while to realize that the ITI and the SOA are not the same, so that there is no contradiction here. Maybe the authors could rephrase the task description somehow to make this less confusing.

- Experimental design: Is there a reasing for using only one fixed block order (slow-fast-slow-fast) without counterbalancing of the starting condition?

- Data analysis: A more finegrained analysis of delta scores might be interesting that separates the first to blocks in halfs. Maybe younger kids already show some adaptation but only in the second half of a block.

- Conclusion: This part is quite long for a conclusion.

6. PLOS authors have the option to publish the peer review history of their article (what does this mean?). If published, this will include your full peer review and any attached files.

Do you want your identity to be public for this peer review? For information about this choice, including consent withdrawal, please see our Privacy Policy.

Reviewer #1: Yes: Katherine Johnson

Reviewer #2: No

---

## [Author Response · Author response to Decision Letter 0]

11 Jan 2023

REVIEWER #1

The aims and the hypotheses were clear. There were a good number of participants in each of the four age groups. […] The task was appropriate and sounded quite fun and engaging for children. The data were treated appropriately. The statistics used were excellent. The discussion was very clearly written. The results were discussed appropriately in light of previous research and theory, and potential links with brain mechanisms were suggested. Figure 3 was particularly helpful.

Are there any limitations the authors would like to note?

RESPONSE: We thank the Reviewer for these very positive comments. One main limitation of the study is the presence of unequal sample sizes per age group, which is now outlined in the manuscript (“One main limitation of this study is the presence of unequal sample sizes per age groups. Therefore, we ensured that every sample size was large enough to afford statistical reliability and modelled the inter-individual variability during data analysis.”, lines 3-5, p. 25). 

REVIEWER #2

Abbreviation ACC for adaptive cognitive control: I recommend against using this abbreviation because it is already a common abbreviation for a brain area (anterior cingulate cortex). This could be confusing for some readers.

RESPONSE: We thank the Reviewer for the suggestion. We have now substituted “ACC” abbreviation using “adaptive CC”. 

Participants: There are very unequal sample sizes per age group (range 35-119). Is there a reason for this?

RESPONSE: Data collection for the different age groups was conducted in different temporal moments. Specifically, only children’s data collection was affected by covid-19 pandemic restrictions, as adults’ data collection was mainly conducted before March 2020. Therefore, we stopped collecting children’s data as soon as every sample size per age group was large enough to ensure statistical reliability, although we did not afford homogeneous groups. To ensure this aspect did not impact our results, we modelled the inter-individual variability in our generalized linear models (1). We have now discussed this limitation in the manuscript (“One main limitation of this study […].”, lines 3-5, p. 25). 

Trial structure: The authors write "The inter-trial-interval (ITI) was randomly manipulated between 200 and 400 ms", but later on it reads "here we used a simplified version only including a short (400 ms) and a long (1,000 ms) SOA interval". It took me a while to realize that the ITI and the SOA are not the same, so that there is no contradiction here. Maybe the authors could rephrase the task description somehow to make this less confusing.

RESPONSE: We thank the Reviewer for the suggestion. We have now rephrased the task description as follows: “The inter-trial-interval (ITI; i.e., the temporal interval between two consecutive trials) was randomly manipulated between 200 and 400 ms.” (lines 7-8, p.9) and “[…] the S1-S2 stimulus onset asynchrony (SOA; i.e., the temporal interval between S1 and S2 within a trial) was varied […]. Unlike the original DTP version (28) here we used a simplified version only including a short (400 ms) or a long (1,000 ms) SOA within each trial, […]” (lines 18-21, p.9).

Experimental design: Is there a reason for using only one fixed block order (slow-fast-slow-fast) without counterbalancing of the starting condition?

RESPONSE: We decided to not counterbalance the starting condition of our fixed block order because, as we state in the manuscript, “[…], the fixed sequence alternating slow-fast-slow-fast blocks was expected to counterbalance the overall speed reduction across the task due to fatigue” (lines 10-12, p.21). Hence, this order allowed disentangling adaptation (increased RTs in slow blocks, reduced RTs in fast blocks) from fatigue (increased RTs in the last block). These two aspects would have been confounded in a fast-slow-fast-slow sequence. In fact, in this latter case, both adaptation and fatigue would be reflected as an increase in RTs in the last slow block. Conversely, the slow-fast-slow-fast sequence allows to disentangle adaptation (reduced RTs in the last fast block) and fatigue (increased RTs in the last fast block). This is further explained in the manuscript (“[…] This ultimately made it possible to compute how efficiently participants succeed in adapting cognitive control using changing global predictability induced bottom-up”, lines 1-3, p.6). We have now better specified this in the manuscript: “This order allowed […]”, (lines 9-12, p.11).

Data analysis: A more finegrained analysis of delta scores might be interesting that separates the first two blocks in halfs. Maybe younger kids already show some adaptation but only in the second half of a block.

RESPONSE: This is an intriguing suggestion, although it could be better addressed with a task purposely designed to assess trial-by-trial learning curves. To answer the Reviewer’s comment, we divided the second block (the Fast1 block) into two halves (Fast1a and Fast1b). Adaptation is assessed as the difference in RTs between fast and slow blocks (Delta = fast - slow); hence, we calculated Delta1a (Fast1a – Slow1) and Delta1b (Fast1b – Slow1). If children show some adaptation at this stage of the task, this should lead to a smaller Delta1b compared to Delta1a, reflecting reduced RTs toward the end of Fast1. However, we found that Delta1a was on average smaller than Delta1b for both young (respectively, 30 ms and 38 ms) and old children (respectively, 30 ms and 42 ms), suggesting that no adaptation occurred during this phase (see Figure 1). Indeed, as can be observed in Figure 2, RTs tend to slightly increase along Fast1, confirming that children do not adapt to the rapid pace of the block. 

Figure 1 (see Response to Reviewers file). Deltas. The picture shows Delta1a (Fast1a – Slow1) and Delta1b (Fast1b – Slow1) for old children (red line) and young children (green line). 

Figure 2 (see Response to Reviewers file). Reaction Times. The picture shows RTs in the Slow1 (first block), Fast1a (first half of the second block) and Fast1b (second half of the second block) for old children (red line) and young children (green line). 

Conclusion: This part is quite long for a conclusion.

RESPONSE: Following the Reviewer’s comment, the Conclusion section has now been shortened.

References

1. Cnaan, A., Laird, N. M., & Slasor, P. (1997). Using the general linear mixed model to analyse unbalanced repeated measures and longitudinal data. Statistics in medicine, 16(20), 2349-2380.

---

## [Decision Letter · Decision Letter 1]

24 Jan 2023

Little fast, little slow, should I stay or should I go? Adapting cognitive control to local-global temporal prediction across typical development

PONE-D-22-33019R1

Dear Dr. TOFFOLI,

We’re pleased to inform you that your manuscript has been judged scientifically suitable for publication and will be formally accepted for publication once it meets all outstanding technical requirements.

Kind regards,

Valentina Bruno

Academic Editor

PLOS ONE

Additional Editor Comments (optional):

Reviewers' comments:

Reviewer's Responses to Questions

**Comments to the Author**

1. If the authors have adequately addressed your comments raised in a previous round of review and you feel that this manuscript is now acceptable for publication, you may indicate that here to bypass the “Comments to the Author” section, enter your conflict of interest statement in the “Confidential to Editor” section, and submit your "Accept" recommendation.

Reviewer #1: All comments have been addressed

Reviewer #2: All comments have been addressed

2. Is the manuscript technically sound, and do the data support the conclusions?

Reviewer #1: No

Reviewer #2: Yes

3. Has the statistical analysis been performed appropriately and rigorously? 

Reviewer #1: Yes

Reviewer #2: Yes

4. Have the authors made all data underlying the findings in their manuscript fully available?

Reviewer #1: Yes

Reviewer #2: Yes

5. Is the manuscript presented in an intelligible fashion and written in standard English?

Reviewer #1: Yes

Reviewer #2: Yes

6. Review Comments to the Author

Reviewer #1: (No Response)

Reviewer #2: (No Response)

7. PLOS authors have the option to publish the peer review history of their article (what does this mean?). If published, this will include your full peer review and any attached files.

Reviewer #1: **Yes: **Katherine Johnson

Reviewer #2: No

---

## [Editor Report · Acceptance letter]

15 Feb 2023

PONE-D-22-33019R1 

Little fast, little slow, should I stay or should I go? Adapting cognitive control to local-global temporal prediction across typical development 

Dear Dr. TOFFOLI:

I'm pleased to inform you that your manuscript has been deemed suitable for publication in PLOS ONE. Congratulations! Your manuscript is now with our production department. 

Kind regards, 

on behalf of

Dr. Valentina Bruno 

Academic Editor

PLOS ONE